# Attitudes towards Anti-SARS-CoV2 Vaccination among Healthcare Workers: Results from a National Survey in Italy

**DOI:** 10.3390/v13030371

**Published:** 2021-02-26

**Authors:** Francesco Di Gennaro, Rita Murri, Francesco Vladimiro Segala, Lorenzo Cerruti, Amina Abdulle, Annalisa Saracino, Davide Fiore Bavaro, Massimo Fantoni

**Affiliations:** 1Clinic of Infectious Diseases, University of Bari, 70121 Bari, Italy; cicciodigennaro@yahoo.it (F.D.G.); annalisa.saracino@uniba.it (A.S.); davidebavaro@gmail.com (D.F.B.); 2Clinic of Infectious Diseases, Catholic University of the Sacred Heart, 00168 Rome, Italy; rita.murri@unicatt.it (R.M.); massimo.fantoni@policlinicogemelli.it (M.F.); 3Dipartimento di Scienze di Laboratorio e Infettivologiche, Fondazione Policlinico Universitario A. Gemelli IRCCS, 00168 Rome, Italy; 4Dipartimento di medicina, University of Padua, 35121 Padua, Italy; lorenzo.cerruti1991@gmail.com; 5Division of Gastroenterology and Hepatology, Città della Salute e della Scienza di Torino Hospital, University of Turin, 10124 Turin, Italy; yesaminabdulle@gmail.com

**Keywords:** COVID-19, vaccine hesitancy, healthcare workers, infectious diseases, cross-sectional survey

## Abstract

Coronavirus disease 2019 (COVID-19) has afflicted tens of millions of people, fostering and unprecedent effort in vaccine development and distribution. Healthcare workers (HCW) play a key role in vaccine promotion and patient guidance, and it is likely that hesitancy among this population will have a major impact on the adoption of a successful immunization policy. To investigate HCW attitudes towards anti-severe acute respiratory syndrome coronavirus 2 (SARS-CoV2) vaccination, we developed an anonymous online cross-sectional survey. 1723 Italian HCW responded. Overall, 1155 (67%) intended to be vaccinated, while 443 (26%) were not sure and 125 (7%) declared refusal. In multivariate analysis, factors associated with hesitancy were using Facebook as the main information source and being a non-physician HCW, while predictors of acceptance included younger age, being in close contact with high-risk groups and having received flu vaccination during the 2019–2020 season. Reasons for hesitancy included lack of trust in vaccine safety (85%) and receiving little (78%) or conflicting (69%) information about vaccines. According to our results, adequate investment in vaccine education for healthcare personnel appears to be urgently needed, prioritizing non-physicians and information quality spread through social media. We hope that our data could help governments and policy-makers to target communication in the ongoing COVID-19 vaccination campaign.

## 1. Introduction

In January 2020, the World Health Organization (WHO) announced the outbreak of a respiratory syndrome caused by a novel coronavirus in the province of Wuhan, China. Since then, the severe acute respiratory syndrome coronavirus 2 (SARS-CoV2) has spread to 189 countries, causing nearly 45 million confirmed cases and more than 1 million deaths [1,2], with no end yet in sight. Healthcare workers, seniors and people with pre-existing conditions are among the categories at greatest risk [3]. So far, non-pharmaceutical interventions (NPIs) have proven effective in suppressing the pandemic [4]—having showed to impact on basic reproduction number (Ro) [5,6]. As a side consequence, however, they contributed to trigger an economic and job crisis worldwide [7]. Furthermore, although effective, mask wearing, social distancing and other NPIs have the paradoxical consequence to prevent the population achieving immunization from the disease, thus allowing the recurrence of additional waves of infection [8].

Still much remains to be learnt about SARS-CoV2 immunity and its duration [9], but it is widely accepted that countries should not rely on the development of natural-occurring immunity as, even with low fatality ratios, an infection-based herd immunity policy would result in substantial mortality worldwide [10]. Against this backdrop, the development of safe and effective vaccines, and the implementation of global and national vaccination programs are the only available tools that humankind can afford to return to pre-pandemic normalcy. Currently, more than 180 vaccines against SARS-CoV2 are in various stages of development [11]. While vaccine development usually takes years to effectively undergo preclinical stages, phase II and phase III trials, the current pandemic and the progress in biotechnologies fostered an unprecedented global effort to make them available at record speed. To date (14 January 2021), in Italy, 908.989 persons received their first vaccine dose, of whom 695.937 were social/health care workers [12].

Since their introduction in the late 18th century [13], few public health initiatives have had a comparable impact on human longevity and health as vaccines. Benefit from vaccination extends beyond prevention of specific infectious diseases, being recognized by the Wolrd Health Organization as an effective measure to promote wealth, economic growth and contrast global inequalities [14]. However, along with increased vaccine use and popularity, there are also public concerns about their safety and efficacy [15]. This loss of confidence, known as “vaccine hesitancy”, involves both vaccinated and non-vaccinated individuals [16], and it has recently been recognized as a major public health challenge by both European Center of Disease Prevention and Control (ECDC) [17] and Italian Ministry of Health [18]. Healthcare workers, even though they play a key role in vaccine promotion and patient guidance, are not immune to vaccine hesitancy, as several studies have shown [19,20,21]. In times of COVID-19 pandemic, at the edge of an unprecedented effort of vaccine development and global distribution, it is likely that vaccine hesitancy among healthcare professionals will have a major impact on the successful adoption of an immunization policy.

In order to advise strategies for vaccination coverage improvement, it is necessary to describe the determinants of vaccine hesitancy in this particular population. From this viewpoint, we conducted the first Italian survey on knowledge, attitudes and practices among healthcare personnel.

## 2. Materials and Methods

### 2.1. Study Design and Setting

A cross-sectional, multicenter survey was conducted from 1 October to 1 November 2020, by administering an anonymous online questionnaire to Italian healthcare workers regardless of the category, setting or region in which they were working.

Ethical approval was granted by the Research Ethics Committee of the Fondazione Policlinico A. Gemelli before study initiation (prot. n 0048483/20, 19 November 2020) Informed consent was requested on the introductory web page prior to survey enrollment.

### 2.2. Questionnaire Development

Development of the questionnaire was informed by a literature review, and it was composed by 26 questions with multiple answers or 5-point Likert-style scale, divided into five sections: (i) demographics and occupation-related information; (ii) attitudes and practice toward anti- SARS-CoV-2 and Flu vaccination; (iii) perceptions on reasons of vaccine hesitancy of the general population; (iv) attitude towards vaccines recommendations; and (v) vaccines relevance’ perception Moreover, an additional section was provided for those declaring that they did not want to be vaccinated, exploring the reasons of personal vaccine hesitancy.

The questionnaire was developed electronically on SurveyMonkey (Survey-Monkey Inc., San Mateo, CA, USA) and distributed over a 4-week period via mailing list and social media.

### 2.3. Statistical Analysis

A descriptive analysis was performed to define the distribution of demographic and occupation-related characteristics of the sample. Vaccine hesitancy was identified as the dependent variable. Vaccine “hesitants” included both those who would not get vaccinated (vaccine resistant/high level of hesitancy) and those being uncertain about their vaccination decision (low levels of hesitancy). Different typologies of postgraduate medical specialties were classified into clinical, non-clinical and surgical areas as for the Italian Ministry of University and Research classification (DM 68/2015).

Independent t test was used to compare groups for continuous variables, whilst a chi-squared test (with the Fisher’s correction if less than five cases were present in a cell) was applied for categorical variables. A logistic regression model was implemented as follows. COVID-19 vaccine hesitancy was considered as a dependent variable and each one of the available factors at the baseline evaluation as independent variables (univariate analysis). In the multivariate analysis factors with a *p*-value < 0.10 by univariate analysis were included.

Multicollinearity among covariates was assessed through the variance inflation factor, taking a value of 2 as cut-off to exclude a covariate. However, no variable was excluded according to this pre-specified criterion. Odds ratios (ORs) as adjusted odds ratios (adj-ORs) with 95% confidence intervals (CIs) were used to measure the strength of the association between factors at the baseline (exposure) and COVID-19 vaccine hesitancy (outcome). All statistical tests were two-tailed and statistical significance was assumed for a *p*-value < 0.05. Statistical analyses were performed with GraphPad Prism 8.0 (GraphPad Software, Inc., San Diego, CA, USA).

## 3. Results

A total of 1841 healthcare workers accessed the survey and 1723 (93.5%) completed all sections and were considered for the analysis. The mean age of respondents was of 35.5 (SD 11.8) years and 920 (53%) were female. Among all healthcare workers participating in the survey, 378 (22%) were health professionals (including nurses and technicians), 337 (20%) were specialized Medical Doctors, 258 (15%) were medical residents (*n* = 437, 41%) and 205 (12%) were from a primary care setting, either General Practitioners (*n* = 135; 8%) or General Practitioner trainees (*n* = 40; 4%), whilst 544 (32%) were medical school graduates with no further specific educational path. Regarding specialists and medical residents, 356 (44%) were from clinical fields, 154 (19%) from non-clinical sectors and 86 (11%) from surgical sectors (Table 1). Most participants (*n* = 1249; 72%) reported to have less than 10 years of length of service in the Health sector. All Italian regions were represented.

Five per cent of participants (*n* = 87) reported to have had a previous infection from SARS-Cov2 or to be currently infected, while 36% (*n* = 636) knew at least one family member or close contact who caught the infection. When asking participants to grade their health status from 0 (very bad) to 10 (excellent) the mean value among all the sample was 9 (SD 1.3), however 275 (16%) participants reported to perceive themselves to be at a higher risk of adverse outcomes due to COVID-19. Sixty-eight percent of respondents (*n* = 1173) reported to have vulnerable persons between their close contacts or cohabitants, in particular people over 65 (*n* = 987; 57%), people with disability/chronic conditions (*n* = 244; 14%) or immunocompromised or in treatment with immunosuppressors (*n* = 220; 13%) (Table 1). As to information sources on SARS-CoV-2 vaccination, participants reported to use, in order: scientific literature (*n* = 1292; 70%), expert opinions (*n* = 851, 49%), webinars and scientific meetings (*n* = 669; 39%), media (*n* = 273; 16%), Facebook groups (*n* = 276; 16%) and journal websites (*n* = 382; 22%). Twenty-four per cent (*n* = 405) of participants declared to be doubtful about their intention towards recommending of anti-SARS-CoV-2 vaccine to their patients, while 6% (*n* = 99) reported that they would not recommend vaccination to their patients.

### 3.1. Attitude and Practice towards Vaccination against Flu and SARS-CoV-2

Forty-seven per cent (*n* = 810) of respondents reported to have been vaccinated against flu during the last flu season (2019–2020) while, when investigating their attitude during the current season (2020–2021), 79% (*n* = 1364) of participants declared their willingness to get flu vaccination (Figure 1).

Instead, as for SARS-CoV-2 vaccination, at the time the questionnaire was administered, 1155 (67%) HCW reported they wanted to get vaccinated as soon as the vaccine would be available, while 26% (443) declared that they still did not know (Figure 2), and 125 (7%) respondents declared vaccine refusal. Overall, 568 (33%) healthcare workers were categorized as vaccine hesitants. Reasons for their personal hesitancy are summarized in Table 2.

### 3.2. Predictors of COVID-19 Vaccine Hesitancy

When comparing COVID-19 vaccine hesitants and not (Table 1 and Table 3), differences (*p*-value < 0.0001) were documented in regards to age group, occupational profile, length of service, having persons at risk among close contacts or cohabitants, information sources on SARS-CoV 2 vaccination, intention towards patients’ COVID-19 vaccine recommendation, flu vaccine undertaken during last season and intention to be vaccinated during the current flu season. Other factors such as sex, geographic or medical area of work, previous infection among family members and close contacts did not have any significant association with vaccine hesitancy.

In the multivariate logistic regression model, being a non-MD health professional (OR 1.82, 95% CI 1.31–2.50) and using Facebook as main information source about anti-SARS-CoV-2 vaccination (OR 1.48, 95% CI 1.06–2.07) remained significantly associated with an increased risk of vaccine hesitancy (*p* < 0.001). On the contrary, being a younger healthcare worker (<30 y) (OR 0.58, 95% CI 0.41–0.83), being in close contact with an high-risk group (OR 0.51, 95% CI 0.35–0.74) and having undertaken seasonal flu vaccine during the 2019–2020 season (OR 0.37, 95% CI 0.29–0.48) resulted to be “protective” towards COVID-19 vaccine hesitancy (Table 3).

To the question “In your opinion, which of the following initiatives would help to increase the number of health workers vaccinating against SARS-CoV-2?”, the answers, in descending order, were increasing information quality about the vaccine (*n* = 682, 40%), and vaccine development (*n* = 486, 28%), implementing an economic incentive (*n* = 367, 21%) and making vaccination mandatory (*n* = 188, 11%) (data not shown). 

### 3.3. Vaccine Hesitancy among the General Population

Participants’ personal view on reasons of vaccine hesitancy in the general population are reported in Figure 3.

### 3.4. Perceptions towards Relevance of Vaccinations in the Fight against COVID-19

The role of seasonal flu vaccination was assessed as extremely and very important respectively by 17% (*n* = 296) and 45% (*n* = 767) of participants, while the role of SARS-CoV-2 vaccination by 48% (*n* = 820) and 35% (*n* = 609) (Figure 4).

## 4. Discussion

In this large survey conducted among 1723 Italian healthcare workers, only 67% (*n* = 1155) reported to be willful to accept COVID-19 vaccination. Twenty-six percent of the participants indicated uncertainty and 125 HCW (7%) declared that they would refuse to be vaccinated. This finding is particularly striking, in our view, considering that the survey was administered at the verge of the Italian “second wave” [22], and that it was administered to professional figures who directly experienced the effects of COVID-19 pandemic [23]. To our knowledge, this is the first study investigating the attitude towards COVID-19 vaccination among healthcare professionals.

Overall, in our study, the main reason of hesitancy was lack of trust in anti-SARS-CoV 2 vaccines with, respectively, 76% and 85% of respondents reporting that their reluctance to get vaccinated was due, at least “to a some extent”, to doubted vaccine efficacy and fear of side effects. These concerns are probably buoyed by a lack of comprehensive, trustable data and media controversy, as suggested by the significant rate of respondents declaring to have received little (78%) or conflicting (69%) information about the vaccines. At the same time, the vast majority of respondents reported mistrust on current containment measures.

Incertitude related to the rapid development process of COVID-19 vaccines was the second most common reason for responding “No” or “Not sure” to intent to be vaccinated, given the high rate of hesitancy attributed to low trust in pharmaceutical companies and control authorities (feared, both, by 58% of respondents). These concerns seem to be specific for the new COVID-19 vaccines and do not apply to vaccination in general, as suggested by the impressively high rate of HCW who reported to have received vaccination against influenza for the year 2020–2021: 79% against 47% for the year 2019–2020. In addition, in the present study, respondents who received vaccination against flu for the 2019–2020 season were significantly more likely to report willingness to get vaccinated against SARS-CoV 2.

Surprisingly, only 16% of respondents reported to feel at higher risk of bad outcomes due to COVID-19, but more than two thirds declared to have, among their cohabitants or close contacts, people belonging to high-risk categories (i.e., seniors, immunocompromised people, people suffering from chronic conditions) and, importantly, this factor resulted significantly associated with willingness to get vaccinated.

At the time this paper is written, vaccines are expected to confer substantial benefit in reducing the overall burden and socio-economical stress related to COVID-19 [9]. In this context, vaccine hesitancy should be taken into careful consideration. With a R0 estimated to be in the range between 2 and 3 [24], current models assume that at least 60% of a given population needs to develop protective immunity in order to extinguish COVID-19 transmission [10]. Compared to other highly-transmissible infectious diseases (i.e., measles), this may appear as a relatively achievable target. We should however not forget that, even when reached, if coverage rates fail to be retained over time, vaccine-preventable infectious diseases are intended to recur [25].

Interestingly, the rate of overall hesitancy towards anti-SARS-CoV2 vaccination reported in our study was similar to the one described in two multinational surveys conducted among the general population [26,27]. This suggests that, in terms of willingness to get vaccinated, healthcare professionals may share the same concerns and may be influenced by the same factors as non-HCW people. Yet, given the essential role that physicians and other HCW play in vaccine promotion [28]—and, more broadly, in public health advising—this is matter of concern. When physicians promote vaccines, they do so knowing that the benefits far outweigh the minimal risks [29], and that each vaccine has been studied extensively to establish its safety profile. However, this might not be perceived today by a consistent part of the HCW population, considered that, in our analysis, the lack of adequate information has been reported as one of the main drivers of hesitancy. Vaccine development and validation has proceeded with impressive results, but maybe not as much attention has been given to the intense wave of uncertainty that this process has spread throughout the world [30]. Nurturing public trust is crucial in contemporary medicine, especially when a disproportionate information demand is not accompanied by a regulated effort in scientific divulgation [31]. Special attention should be reserved to non-physicians HCW and to prioritizing the quality of information spread through social media as, in our study, both these factors were associated to an increased risk to report vaccine hesitancy. In this view, sufficient investment in vaccine education for healthcare personnel appears to be urgently needed [32].

Our study has several limitations. First, respondent selection was not randomized, thus not allowing our results to be generalized to the whole Italian HCW population due to the possibility of a selection bias. In particular, non-physicians HCW and people above 50 years old were likely underrepresented. Another obvious limitation of this study is that the rate of willingness to be vaccinated, and the underlying reasons sustaining hesitancy do not apply to the general population, as it falls outside the study aim. Furthermore, reporting one’s intent to be vaccinated might not necessarily mean that this individual will get vaccinated, as vaccine decision are multi-layered and considerably change over time [29]. In fact, as with other similar cross-sectional surveys, it must be remembered that our study represents a single snapshot in a multifaced and extremely dynamic panorama, in which evidence and perceptions may vary on a day-to-day base. In the two months since the survey administration started, for instance, several highly impactful events may have shaped public opinion and vaccine perception, such as the publication of the first publicly available data from a phase III trial, and the first person receiving a clinically authorized, fully tested coronavirus vaccine shot.

Results from anti-SARS-CoV2 vaccines phase III trials have outlined [33] that new technologies, when accompanied by a sufficient investment of resources, are capable of developing a safe and effective tool against emergent infectious diseases in less than a year. However, if we want these efforts to be converted in a sustained protection able to allow us to recover from the loss of health, resources and lives associated with the ongoing pandemic, it may be important to recognize the determinants of hesitancy and to provide a comprehensive and accessible information both for HCW and the general population.

## 5. Conclusions

Given the need to achieve and keep the world population above herd immunity threshold for COVID-19, a continuous effort is going to be required from public health authorities to preserve trust and erode vaccine hesitancy. In this fight, healthcare workers will always play a major role, as they will keep being both a high-risk population and the first source to which patients will seek advice. We hope that our data could help governments, public health professionals and policy-makers to target educational interventions and communication in the upcoming COVID-19 vaccination campaign.

## Figures and Tables

**Figure 1 viruses-13-00371-f001:**
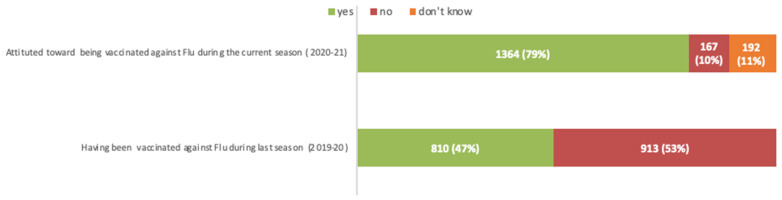
Attitude towards vaccination against seasonal Flu: comparison between the current (2020–2021) and the last flu season (2019–2020).

**Figure 2 viruses-13-00371-f002:**
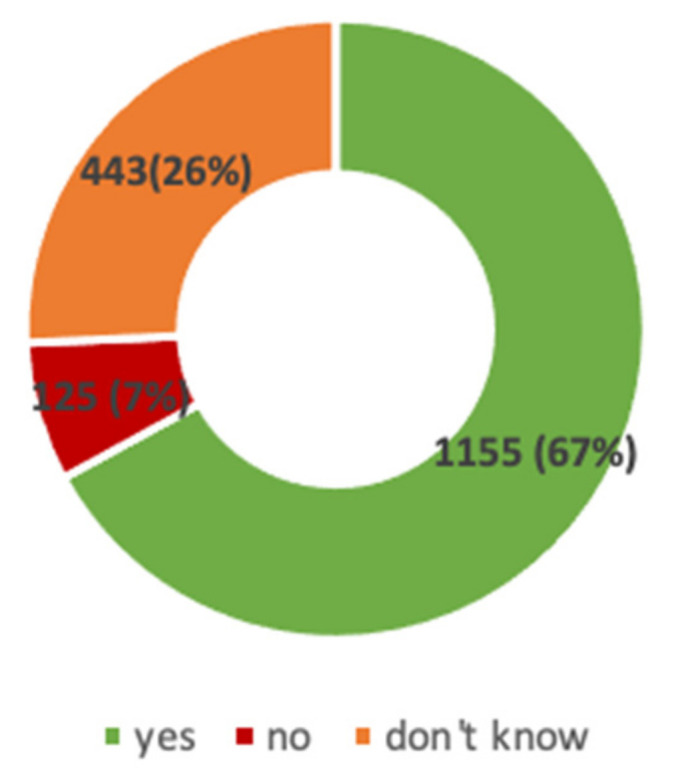
Attitude towards vaccination against Sars-CoV-2.

**Figure 3 viruses-13-00371-f003:**
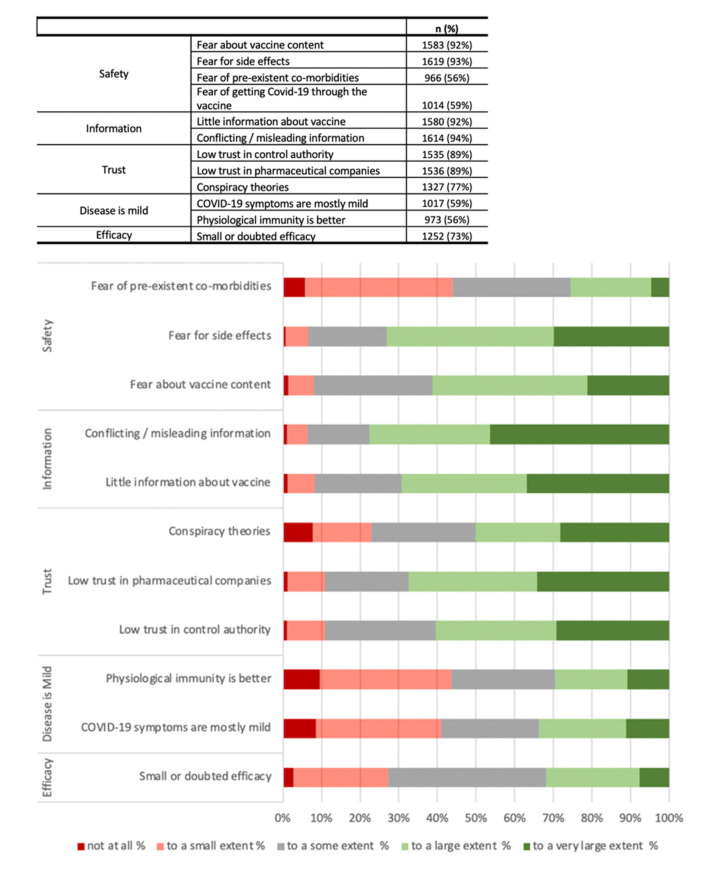
Participants’ opinions on reasons of vaccine hesitancy among the general population.

**Figure 4 viruses-13-00371-f004:**
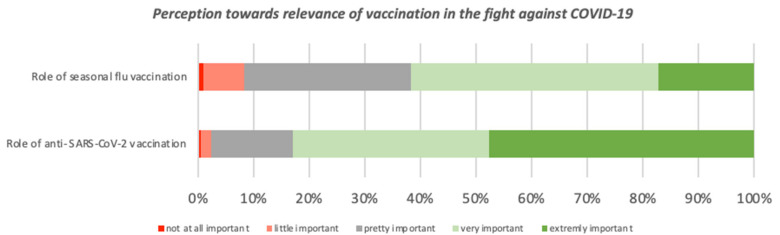
Perception of 1723 participants towards relevance of vaccinations in the fight against COVID-19.

**Table 1 viruses-13-00371-t001:** Characteristics of the 1723 participants compared by being anti-SARS CoV2 vaccine hesitant or not.

Characteristic		Total Participants (*n* = 1723)	Intent to Be Vaccinated		*p*-Value
			No or not Sure (*n* = 568)	Yes (*n* = 1155)	
Sex (*n*, %)	Female	920 (53)	303 (33)	617 (67)	0.999
	Male	803 (47)	265 (33)	538 (67)	
Age group (*n*, %)	<30	762 (44)	184 (24)	578 (76)	**<0.0001**
	30–40	527 (31)	205 (39)	322 (61)	
	41–50	200 (12)	107 (54)	93 (47)	
	51–60	106 (6)	47 (44)	59 (56)	
	>60 years	128 (7)	25 (20)	103 (80)	
Geografic Area (*n*, %)	Central Italy	385 (22)	119 (31)	266 (69)	0.2443
	Northern Italy	622 (36)	197 (32)	425 (68)	
	Southern Italy and Islands	716 (42)	252 (35)	464 (65)	
Occupational profile (*n*, %)	Specialised Medical Doctor	337 (20)	103 (31)	234 (69)	**<0.0001**
	Medical resident	259 (15)	55 (21)	204 (79)	
	Medical Doctor	544 (32)	131 (24)	413 (76)	
	General Practitioner	135 (8)	49 (37)	85 (63)	
	GP trainee	70 (4)	16 (23)	54 (77)	
	Non-MD health professional	378 (22)	214 (57)	164 (43)	
Area of work (*n*, %)	Surgical	86 (11)	26 (30)	60 (70)	0.3731
	Clinical	356 (44)	95 (27)	261 (73)	
	Non-clinical	154 (19)	37 (24)	117 (76)	
	Primary health care	205 (26)	65 (32)	140 (68)	
Length of service (*n*, %)	<10 y	1249 (72)	364 (29)	885 (71)	**<0.0001**
	>10 y	474 (28)	204 (43)	270 (57)	
Self-perceived risk (*n*, %)	Previous or current SARS-CoV-2 infection	87 (5)	33 (38)	54 (62)	0.3491
	Previous or current SARS-CoV-2 Infection in family members or close contacts	626 (36)	213 (34)	413 (66)	0.4888
	Rating self-perceived health status (mean, SD)	8.5 (1.3)	8.4 (1.4)	8.68 (1.3)	-
	Self-perceived higher risk of contagion or bad outcome for COVID-19 due to health status	275 (16)	106 (19)	169 (15)	**0.0357**
People at risk between close contacts or cohabitants (*n*, %)	Over > 65	987 (57)	299 (30)	688 (70)	**0.0071**
	Children < 12	440 (26)	166 (38)	274 (62)	
	People with disability or current serious disease	244 (14)	72 (30)	172 (70)	
	Immunocompromised or in treatment with immunosuppressors	220 (13)	57 (26)	162 (74)	
**Trust in Current Containment Measures (Mean, SD)**		6 (1.9)	5.8 (1.9)	6.07 (1.9)	-
Information sources on SARS-CoV-2 vaccination (*n*, %)	Scientific Literature	1202 (70)	335 (28)	867 (72)	**<0.0001**
	Expert opinions	851 (49)	275 (33)	569 (67)	
	Scientific meeting	669 (39)	186 (28)	485 (72)	
	Media	273 (16)	65 (24)	208 (76)	
	Facebook group	276 (16)	120 (43)	156 (57)	
	Journal and website	382 (22)	132 (35)	250 (65)	
Attitude towards patients recommendation (*n*, %)	would recomment	1219 (71)	17 (20)	1046 (91)	**<0.0001**
	uncertain	405 (24)	32 (57)	84 (7)	
	would not recommend	99 (6)	74 (13)	25 (2)	

*p*-values < 0.05 are presented in boldface.

**Table 2 viruses-13-00371-t002:** Reasons of personal COVID-19 vaccine hesitancy among the 568 hesitant participants.

Self-Reported Reasons of Vaccine Hesitancy		*n* (%)
Safety	Fear about vaccine content	455 (80%)
	Fear for side effects	482 (85%)
	Fear of pre-existent co-morbidities	253 (45%)
	Fear of getting Covid-19 through the vaccine	234 (41%)
Information	Little information about vaccine	441 (78%)
	Conflicting/misleading information	390 (69%)
Trust	Low trust in control authority	332 (58%)
	Low trust in pharmaceutical companies	329 (58%)
	Conspiracy theories	75 (13%)
Complacency	COVID-19 symptoms are mostly mild	117 (21%)
	Physiological immunity is better	148 (26%)
Efficacy	Small or doubted efficacy	433 (76%)

**Table 3 viruses-13-00371-t003:** Predictors of COVID-19 vaccine hesitancy.

		Univariate Analysis *		Multivariate Analysis °	
		OR (95% CI)	*p*-Value	aOR (95%CI)	*p*-Value
**Sex**	**Male**	0.99 (0.82–1.21)	0.999	1.02 (0.79–1.32)	0.19
Age group	<30 y	0.47 (0.39–0.58)	**<0.0001**	0.58 (0.41–0.83)	**<0.0001**
Geografic Area	Northen Italy	0.91 (0.74–1.12)	0.3443	0.95 (0.73–1.24)	0.46
Occupational profile	Health Professionals	3.65 (2.87–4.63)	**<0.0001**	1.82 (1.31–2.50)	**<0.0001**
Area of work	Clinical	1.454 (1.11–1.88)	**0.004**	1.072 (0.76–1.48)	0.06
Length of service	>10 y	1.83 (1.47–2.28)	**<0.0001**	1.32 (0.82–2.19)	0.02
Previous or current SARS-CoV-2 infection		1.25 (0.81–1.97)	0.3491	1.22 (0.68–2.13)	0.53
Previous or current SARS-CoV-2 Infection in family members or close contacts		1.08 (0.87–1.32)	0.888	0.99 (0.76–1.29)	0.02
Rating self-perceived health status (mean, SD)		0.81 (0.83–1.32)	0.03	0.99 (0.89–1.10)	0.02
Information sources on SARS-CoV 2	Facebook group	1.71 (1.3–2.22)	**<0.0001**	**1.48 (1.06–2.07)**	**<0.0001**
Seasonal flu vaccination	Flu vaccination during last season	0.28 (0.22–0.35)	**<0.0001**	**0.37 (0.29–0.48)**	**<0.0001**
	Intention towards Flu vaccination for the current season	0.66 (0.43–0.98)	**<0.0001**	0.79 (0.61–1.02)	0.02

OR: odds ratio; aOR: adjusted odds ratio; *: Chi-squared test; °: Multivariate logistic regression; NA: not applicable. *p*-values < 0.05 are presented in boldface.

## Data Availability

All data used in this study are available upon reasonable request to the corresponding author.

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
