# Peer review of "Attitudes towards Anti-SARS-CoV2 Vaccination among Healthcare Workers: Results from a National Survey in Italy"

_viruses, 2021, doi:10.3390/v13030371_

Round 1

Reviewer 1 Report

Dear Editor,

the paper entitled " Attitudes towards anti-SARS-CoV2 vaccination among healthcare workers: results from a national survey in Italy" is an interesting study that deals with the knowledge, attitudes and practices concerning healthcare personnel towards anti-SARS-CoV2 vaccination. The manuscript is well written and well supported by consistent references. The conclusions are largely sound and help to reduce the limited existing knowledge. Therefore, I think that the paper has enough quality to be published in Viruses.

Author Response

To Viruses Editor,

We have appreciated the positive feedback to our manuscript “Attitudes towards anti-SARS-CoV2 vaccination among healthcare workers: results from a national survey in Italy”.

We are very grateful to all the useful suggestions made by the referees and we have implemented  them in the text. We have also satisfied the technical requirements according to the journal guidelines. Modifications have been highlighted using the "track changes" feature. Also, as requested, a native English speaker has been engaged to improve the fluency and the readability of the manuscript.

We believe that the revision proposed by the referees, and further implemented in the text, contributed to improve the manuscript. Thus, we kindly ask you to re-consider the manuscript for publication.

Please find a point-by-point response to the referees’ comments below.

Best regards,

Dr. Francesco Vladimiro Segala

Reviewer's Responses to Questions

Reviewer 1

The paper entitled " Attitudes towards anti-SARS-CoV2 vaccination among healthcare workers: results from a national survey in Italy" is an interesting study that deals with the knowledge, attitudes and practices concerning healthcare personnel towards anti-SARS-CoV2 vaccination. The manuscript is well written and well supported by consistent references. The conclusions are largely sound and help to reduce the limited existing knowledge. Therefore, I think that the paper has enough quality to be published in Viruses.

Response:

We thank you very much for the encouraging feedback on our manuscript. Healthcare workers play a key role in vaccine promotion and patient guidance, and it is likely that hesitancy among this population will have a major impact on the adoption of a successful immunization policy.

Reviewer 2 Report

The manuscript by Gennaro et al., was an interesting study on the potential effects of vaccination related health communications. The authors rationally designed the investigations and performed certain analysis. It was understood that the study was potentially useful but with little scientific novelty, regarding the viruses, which might be more suitable for more specific journals. Main points:

  • The authors should improve the methodology to increase the novelty of the study.
  • Line 311 and related analysis “recognize the uncertainty” needs to be explained with details and assessed. List the factors of uncertainty.
  • Section 5. Conclusions, Lines 315-321 were not supported by data. The writing like a newspaper instead of a scientific paper.

Author Response

To Viruses Editor,

We have appreciated the positive feedback to our manuscript “Attitudes towards anti-SARS-CoV2 vaccination among healthcare workers: results from a national survey in Italy”.

We are very grateful to all the useful suggestions made by the referees and we have implemented  them in the text. We have also satisfied the technical requirements according to the journal guidelines. Modifications have been highlighted using the "track changes" feature. Also, as requested, a native English speaker has been engaged to improve the fluency and the readability of the manuscript.

We believe that the revision proposed by the referees, and further implemented in the text, contributed to improve the manuscript. Thus, we kindly ask you to re-consider the manuscript for publication.

Please find a point-by-point response to the referees’ comments below.

Best regards,

Dr. Francesco Vladimiro Segala

Reviewer's Responses to Questions

Reviewer 2

The manuscript by Gennaro et al., was an interesting study on the potential effects of vaccination related health communications. The authors rationally designed the investigations and performed certain analysis. It was understood that the study was potentially useful but with little scientific novelty, regarding the viruses, which might be more suitable for more specific journals. Main points:

  • The authors should improve the methodology to increase the novelty of the study.
  • Line 311 and related analysis “recognize the uncertainty” needs to be explained with details and assessed. List the factors of uncertainty.
  • Section 5. Conclusions, Lines 315-321 were not supported by data. The writing like a newspaper instead of a scientific paper.

Response:

We thank you very much for the encouraging feedback on our manuscript.

Concerning the scientific novelty, we believe that an accurate description of vaccinal attitudes among healthcare workers could provide useful information to identify the determinants of vaccine hesitancy and guide interventions. In regard to the interest for publication in Viruses, we think that our article may suit the journal special issue “Viruses and Vaccines: Past Successes and New Approaches” as it focuses on “elimination […] by overcoming the challenges of timely delivery to key populations, including […] fight against vaccine hesitancy”.

Furthermore, in the revised version:

  • In the line 311, we replaced the term “uncertainty” with “hesitancy”, as the first could sound misleading to the reader. We are grateful to reviewer 2 for this observation.

Below, the corrected version: “Results from anti-SARS-CoV2 vaccines phase III trials have outlined [35] that new technologies, when accompanied by a sufficient investment of resources, are capable of developing a safe and effective tool against emergent infectious diseases in less than a year. However, if we want these efforts to be converted in a sustained protection able to allow us to recover from the loss of health, resources and lives associated with the ongoing pandemic, it may be important to recognize the determinants of hesitancy and to provide a comprehensive and accessible information both for HCW and the general population”.

  • We modified the conclusion, following reviewer suggestions. Below, the revised version: “Given the need to achieve and keep the world population above herd immunity threshold for COVID-19, a continuous effort is going to be required from public health authorities to preserve trust and erode vaccine hesitancy. In this fight, healthcare workers will always play a major role, as they will keep being both an high-risk population and the first source to which patients will seek advice. We hope that our data could help governments, public health professionals and policy-makers to target educational interventions and communication in the upcoming COVID-19 vaccination campaign”.

Round 2

Reviewer 2 Report

The manuscript entitled by “Attitudes towards anti-SARS-CoV2 vaccination among 2 healthcare workers: results from a national survey in Italy” is a comprehensive research article on the potential roles of vaccine knowledge and personal attitude on the vaccination decision making. The authors have focused on the healthcare workers and made appropriate revisions according to previous comments. Minor modifications needed: Page 2 Lines 45-47 need references and to be based on scientific data. Table 1 needs reorganizing/remodeling. Table 4 and related experimental design, Lines 207-209, rational design was unknown.

Author Response

Reviewer 2

The manuscript entitled by “Attitudes towards anti-SARS-CoV2 vaccination among healthcare workers: results from a national survey in Italy” is a comprehensive research article on the potential roles of vaccine knowledge and personal attitude on the vaccination decision making. The authors have focused on the healthcare workers and made appropriate revisions according to previous comments. Minor modifications needed: Page 2 Lines 45-47 need references and to be based on scientific data. Table 1 needs reorganizing/remodeling. Table 4 and related experimental design, Lines 207-209, rational design was unknown

Response:

We thank you very much for the encouraging feedback on our manuscript. All suggestions have been implemented in the new version of the article, and modifications have been highlighted using the "track changes" feature. More in detail:

  • Page 2, Lines 45-47. We added further references and we modified the text as follows: “Still much remains to be learnt about SARS-CoV2 immunity and its duration [9], but it is widely accepted that countries should not rely on the development of natural-occurring immunity as, even with low fatality ratios, an infection-based herd immunity policy would result in substantial mortality worldwide [10].” Bibliography section has been revised accordingly.
  • Table 1 has been extensively reorganized. Please find the modified version in the revised manuscript
  • After careful deliberation with all authors, as requested by reviewer 2, we decided to remove Table 4 and related Lines 207-209 from the manuscript.

We are thankful for all the issued highlighted by reviewers. Hence, we believe that reviewer’s suggestions substantially improved the quality of our work.

Kind regards,

Dr Francesco Vladimiro Segala
